# Experiences of immigrants when accessing mental health services and psychosocial supports in Canada: Protocol for a scoping review

Temitayo Sodunke[1], Gerald Agyapong-Opoku[1], Omolayo Anjorin[2], Belinda Agyapong[3], Mutiat Sulyman[4], Sanjana Sridharan[4], Somtoo Rita Henry[5], Ejemai Eboreime[2,4]*

1 School of Health and Human Performance, Faculty of Health, Dalhousie University, Halifax, Nova Scotia, Canada, 2 Department of Psychiatry, Faculty of Medicine, Dalhousie University, Halifax, Nova Scotia, Canada, 3 Department of Psychiatry, Faculty of Medicine & Dentistry, University of Alberta, Edmonton, Alberta, Canada, 4 Department of Psychiatry, Nova Scotia Health Authority, Halifax, Nova Scotia, Canada, 5 Faculty of Basic Medical Science, College of Health Sciences, Nnamdi Azikiwe University, Awka, Nigeria

* ejemai.eboreime@dal.ca

## Abstract

In Canada, the mental health needs of immigrants from diverse ethnic backgrounds are a growing concern, as they are at a higher risk for adverse mental health outcomes compared to native-born Canadians. These challenges are often driven by stressors experienced both before and after their settlement. The limited research on immigrant experiences in accessing mental health and psychosocial support highlights the need for our study. Compounding this gap, existing data reveal significant challenges immigrants face in accessing mental health services. Therefore, to better understand and address the needs of Canada's immigrant population, we are conducting a scoping review to identify and synthesize the existing literature on immigrants' experiences when accessing mental health services and psychosocial support. This scoping review will follow the methodological framework outlined by Arksey and O'Malley and complimented by Joanna Briggs Institute guidelines, as well as the Preferred Reporting Items for Systematic Reviews and Meta-Analyses (PRISMA) extension for scoping review. To ensure intercoder reliability (ICR), the team will conduct a calibration exercise before initiating the screening process. This will involve an independent screening by all reviewers followed by a team discussion to resolve any discrepancies and clarify the application of inclusion/exclusion criteria. All stages of the selection process will be conducted using Covidence, a web-based software platform designed to support systematic reviews. Our analysis will involve both quantitative and qualitative approaches to align with the nature of the included articles and address our research questions. By synthesizing the current state of knowledge, identifying gaps, and highlighting potential solutions, this protocol aims to contribute to the ongoing crucial efforts to improve mental health outcomes for Canada's immigrant population.

**Data availability statement:** All relevant data are within the paper and its Supporting Information files.

**Funding:** Eboreime E. This work was funded by the QEII Foundation Translating Research into Care (TRIC) Award 1029792. The funders played no role in the study design, data collection, data extraction, results, and discussion. https://qe2foundation.ca/qeii-health-sciences-centre-staff/grants-health-centre-staff-physicians/translating-research-care-tric-grant-program

**Competing interests:** No authors have competing interests.

# Introduction

Recent trends have identified globalization, political instability, climate change, and advancement in transportation and communication technologies as drivers of the heightened migration patterns of many individuals across the globe [1]. In Canada, over 200,000 immigrants are welcomed annually, and their major roots stem from the Asian and African continents [2,3]. According to reports, the mental health needs of immigrants from diverse backgrounds remain concerning as they are at a higher risk of adverse mental health outcomes often caused by stressors before, during and after their settlement journey [2,4]. A noteworthy concept often termed the 'healthy immigrant effect' is a view held by most scholars. It is believed that although immigrants arrive in Canada with a higher level of physical and mental health status compared with native-born Canadians, after immigration, this state of health declines rapidly depending on the contextual stressors and support available for this group, and this includes mental health [1,2,5,6]. A review of four major national health surveys, from the 1996 NPHS Cycle 2 to the 2005 CCHS Cycle 3.1, the healthy immigrant effect was evident in data showing consistently lower rates of depression, alcohol dependence, and self-rated mental health issues, as compared to the general population [6,7]. Furthermore, this deterioration in health status has been attributable to immigration and settlement stressors as well as experiences of racialization and discrimination among these immigrants [2,8]. Research indicates that various barriers—including religion, language, geographic location, financial insecurity, systemic discrimination, racism, and cultural or social factors—are associated with multiple challenges. These challenges result in limited access to mental healthcare compared to other Canadians, largely due to a lack of cultural competency among mental health providers, as well as stigma and misconceptions about mental health within immigrant communities [1,2,9,10]. As the WHO (2014) emphasizes, racialized immigrants in Canada face compounded disadvantages due to barriers in accessing mental health services. These barriers, including language, discrimination, and lack of culturally competent care, significantly hinder their ability to seek or receive appropriate support, thus placing them at a greater disadvantage compared to other Canadians [2,10,11]. Furthermore, the need for accessible, culturally sensitive care is critical across all age groups to ensure the well-being of this diverse population, as mental health is a cornerstone of overall health and integration into Canadian society [2]. Without equitable access to such services, immigrants are disproportionately affected by unmet mental health needs, underscoring the importance of addressing these gaps for improved health outcomes.

Similarly, improving access to mental health services for diverse communities is a key component of the Mental Health Commission's 2016 strategic direction. Meeting the needs of Immigrant, Refugee, Ethnocultural, and Racialized (IRER) populations has become an urgent priority for Canada's mental health system and its service providers [5]. However, immigrants from diverse ethnic groups across the country still face significant challenges in accessing mental health services [5]. For example, the detrimental effect of mental ill health on immigrants have been associated with difficulties around unemployment, depression, precarious employment conditions, linguistic barriers, low self-esteem [12,13]. Additional difficulties include navigating the Canadian education system (e.g., different learning styles and approaches, bullying and inability to build friendships,) family tensions, income security among others [12,13].

Research suggests that limited access to mental healthcare is another likely explanation for how new immigrants in Canada perceive mental health and their use of related services [2,8,14,15]. This has resulted from barriers like stigma around mental illness, language barriers, cultural interpretations of the condition, lack of knowledge on existing healthcare services, and individual's understanding of accessing services [1,16]. In corroboration with earlier

viewpoints on the hindrances that could pose a threat to immigrants' access and utilization of mental health services, cultural and religious beliefs, as well as a stigma by these individuals themselves, tend to influence their mental health seeking behaviour [16–18]. The most obvious theme to emerge from these findings demonstrates that a standardized, uniform, and non-targeted approach to mental health care is often inadequate. Such approaches, which lack cultural competence, may fail to address the diverse needs of linguistically and ethnically varied immigrant groups in Canada. Interestingly, immigrants' perspectives on mental health, cultural and religious beliefs, and stigma are often identifiable as influential to their mental health seeking behaviour (Chaze et al., 2015; Gopalkrishnan, 2018). Borque's (2011) study also emphasizes that even though this population is resilient, extremely resourceful, and highly adaptable in the face of adversity, poor mental health outcomes have been documented for them globally [19]. Therefore, to better address the needs of the immigrant population within Canada, this scoping review is set to map out the scope of literature available on the experiences of immigrants while accessing mental health services and psychosocial support that is obtainable.

## Justification statement

This review holds particular significance as it draws on the lived experiences of its authors, all of whom are landed immigrants working within the healthcare sector. Despite their professional expertise and familiarity with the Canadian healthcare system, they continue to face substantial barriers in accessing mental health care.

Canada has a rich history of immigration, with millions of individuals from across the globe choosing the country as their new home. As of 2021, over 8.3 million people—approximately one-quarter (23.0%) of the population—were or had ever been landed immigrants or permanent residents, marking the highest proportion since Confederation [20]. This figure surpassed the previous 1921 record of 22.3% and represents the largest immigrant population among the G7 nations [20]. Given that 23% of Canada's population is comprised of immigrants and continues to grow, it is essential to examine the unique barriers and facilitators that affect their access to mental health care [20].

While the healthcare system aims for equity and inclusivity, the ongoing challenges faced by immigrants reveal critical gaps that need to be addressed. This review aims to synthesize existing literature and offer evidence-based recommendations for developing a more equitable and culturally responsive mental health system that meets the needs of Canada's diverse population.

### Research Questions

We aim to answer the following questions:

1. What are the experiences of immigrants when accessing mental health services and psychosocial support in Canada?

2. What are the barriers and enablers for access to mental health services and psychosocial support by immigrants in Canada?

3. What recommendations have been made in the literature for improving access to and experiences with mental health services and psychosocial support for immigrants in Canada?

## Methods

This scoping review will follow the methodological framework outlined by Arksey and O'Malley [21] and complimented with Joanna Briggs Institute, as well as the Preferred Reporting

Items for Systematic Reviews and Meta-Analyses (PRISMA) extension for scoping reviews see Prisma P S1 Table [22–24]. The five stages of Arksey and O'Malley's approach will guide our process: identifying the research questions, identifying relevant articles, study selection, charting the data, and collating, summarizing, and reporting the results [21].

## Search strategy

We will conduct a comprehensive search of the following electronic databases: MEDLINE (Ovid MEDLINE ALL), EMBASE (Ovid interface), CINAHL (EBSCOhost interface), PsycINFO (Ovid), Social Work Abstracts (EBSCOhost), SocINDEX (EBSCOhost), Web of Science Core Collection, and Scopus (Elsevier). These databases were selected to ensure comprehensive coverage of medical, health, nursing, allied health, and broader scientific fields relevant to our research questions.

The search strategy will be developed in consultation with a health sciences librarian and will include a combination of subject headings and keywords related to the following concepts: immigrants, mental health services, psychosocial support, and Canada. The search strategy is developed and will be implemented by the authors. The search will be limited to English-language peer-reviewed and no timeframe.

A draft search strategy developed by the authors and in consultation with the health librarian for MEDLINE is provided S2 Table. This strategy will be adapted for use in other databases S3, with database-specific subject headings used where available.

## Study selection

### Inclusion criteria

1. Articles will be eligible for inclusion if they meet the following criteria:

2. Focus on first generational immigrants- refugees, newcomers (permanent residents (including people who have received "approval-in-principle" from Immigration, Refugees and Citizenship Canada to stay in Canada)) [25], refugees (protected persons), temporary residents (including student, worker, or temporary resident permit holders, or asylum seekers living in Canada

3. Address access to mental health services and/or psychosocial support

4. Evaluate barriers and/or enablers to accessing these services

5. Are original peer-reviewed journal articles using quantitative, qualitative, or mixed methods, case studies and grey literature

6. Are specific to the Canadian context

### Exclusion criteria

We will exclude:

1. Systematic reviews, meta-analyses, interventional articles, commentaries, editorials, opinion pieces

2. Articles that do not focus on mental health conditions

3. Articles with non-immigrant or local migration (in-country) populations

4. Non-peer-reviewed articles, graduate student theses, and conference reports

The study selection process will be conducted in two stages:

1. Title and Abstract Screening: Three independent reviewers will screen all titles and abstracts against the inclusion and exclusion criteria. Any disagreements will be resolved through discussion, with a fourth reviewer consulted if necessary.

2. Full-Text Review: Two independent reviewers will assess the full texts of potentially eligible articles. Disagreements will be resolved through discussion or consultation with a third reviewer.

To ensure intercoder reliability (ICR), the team will conduct a calibration exercise before initiating the screening process. This will involve all reviewers independently coding a random sample of 50 titles and abstracts as a pilot test. Intercoder reliability will be assessed using an appropriate ICR metric (e.g., Cohen's kappa or Krippendorff's alpha), followed by a team discussion to resolve discrepancies and refine the application of inclusion/exclusion criteria [26].

All stages of the selection process will be conducted using Covidence, a web-based software platform designed to support systematic reviews [27]. This will facilitate the removal of duplicates and the independent screening of titles, abstracts, and full texts.

## Data extraction

A standardized data extraction form will be developed in Microsoft Excel to capture relevant information from the included articles. Two reviewers will independently extract data from each included study. The extraction form will capture the following information:

1. Study characteristics (e.g., authors, publication year, study design, objectives)

2. Population characteristics (e.g., sample size, age range, gender, ethnicity, immigration status)

3. Mental health characteristics (e.g., specific conditions, comorbidities)

4. Service access and utilization details

5. Key findings related to experiences, barriers, and enablers of access

6. Recommendations for improving access and experiences

The complete data extraction form is provided as S1 Table

To ensure intercoder reliability (ICR) during data extraction, a random sample of 20% of the included articles will be independently coded by both reviewers. Intercoder agreement will be assessed using Cohen's kappa coefficient, with a target reliability threshold of 0.8 or higher. Any discrepancies will be resolved through discussion or, if needed, consultation with a third reviewer

## Data analysis

Our analysis will involve both quantitative and qualitative approaches to align with the nature of the included articles and address our research questions.

**Quantitative Analysis:** We will use descriptive statistics to summarize the characteristics of the included articles, such as publication year, study design, sample size, and geographic location. Tables and charts will be created to visually represent the distribution of articles across key variables, including types of mental health services accessed, prevalence of specific barriers and enablers, and frequency of different recommendations.

**Qualitative Analysis:** We will use thematic analysis to synthesize findings from qualitative articles and the qualitative components of mixed-methods articles [28]. This will involve an

iterative process of coding the data, identifying patterns and themes, and developing a thematic framework that captures key aspects of immigrants' experiences, barriers and enablers to access, and recommendations for improvement. We will follow Braun and Clarke's six-step process for thematic analysis (23).

**Integration of Findings:** We will integrate the quantitative and qualitative findings to provide an equity-focused synthesis of the literature. This process will involve identifying convergent and divergent findings and developing a conceptual framework that highlights the systemic and contextual factors shaping immigrants' equitable access to and experiences with mental health services and psychosocial supports in Canada. The framework will address systemic factors like health policies, healthcare infrastructure, and legal barriers to service access. It will also explore contextual factors such as cultural competence, social support networks, and language barriers. Psychosocial and cultural factors, including acculturation stress, stigma, and trauma, will be considered, along with equity issues related to disparities in access and outcomes for immigrants, especially those from marginalized groups. Lastly, we will synthesize the quantitative and qualitative findings to provide a comprehensive understanding of barriers and solutions to improve immigrants' equitable access to mental health services.

Throughout the analysis process, regular meetings will be held among team members to discuss progress, resolve discrepancies, and ensure consistency in the application of analysis methods. We will keep detailed records of our analysis process, including coding schemes and theme development, to ensure transparency and reproducibility.

**Consultation with Stakeholders:** To enhance the validity and relevance of our findings, we will consult with specific key stakeholders, including immigrant community leaders (e.g., heads of local immigrant organizations), mental health service providers (e.g., clinicians, counselors, social workers working with immigrant populations), and policymakers (e.g., public health officials and government representatives at the municipal and provincial levels). The consultation will take place after the initial analysis of findings and will involve presenting a summary of preliminary results to stakeholders for their feedback. This consultation will be conducted online through Microsoft Teams and Outlook to increase accessibility, ensuring that stakeholders from diverse geographic locations can participate. Their input will be gathered through structured discussions and surveys, allowing them to share insights and experiences that will inform the refinement of our findings and contribute to the development of targeted recommendations for practice and polices

## Ethics and dissemination

As this scoping review will only include published literature and will not involve human participants, ethics approval is not required. The results of this review will be disseminated through publication in a peer-reviewed journal, presentation at relevant conferences, and sharing with key stakeholders in the immigrant health and mental health sectors.

## Results

The results of this scoping review will be presented in accordance with the PRISMA-ScR guidelines (20,21), offering a comprehensive overview of the literature on immigrants' experiences with mental health services and psychosocial supports in Canada. Our presentation will be structured to provide a clear and detailed synthesis of the findings.

We will begin by describing the study selection process through a PRISMA flow diagram, which will illustrate the number of articles identified, screened, assessed for eligibility, and ultimately included in the review. This visual representation will be accompanied by a

narrative description of the search results, including the total number of articles screened and the primary reasons for exclusion at the full-text review stage.

Following this, we will present a detailed descriptive summary of the characteristics of included articles. This summary will encompass publication trends over the 2014–2024 period, the distribution of study designs, and the geographic representation across Canadian provinces and territories. We will also describe the range of sample sizes and participant characteristics, including age ranges, gender distribution, countries of origin, immigration statuses, and length of time in Canada. Additionally, we will outline the types of mental health services and psychosocial supports addressed in the literature. To enhance clarity and highlight patterns, this information will be synthesized in tables and visualized through charts and graphs.

Our analysis of immigrants' experiences will be presented thematically, focusing on access pathways to mental health services, the quality and cultural appropriateness of services received, interactions with healthcare providers, and the impact of services on mental health outcomes. To provide rich, contextual data, we will incorporate illustrative quotes from qualitative articles. Any quantitative measures of experiences, such as satisfaction ratings or service utilization rates, will be summarized and, where possible, presented in comparative tables or graphs.

The review will then delve into a comprehensive analysis of barriers and enablers to accessing mental health services and psychosocial supports. We will describe the prevalence of specific barriers and enablers across articles, categorizing these factors into themes such as cultural, linguistic, socioeconomic, and structural. Our analysis will explore how these barriers and enablers differ across immigrant subgroups or types of mental health services, as well as examining the intersectionality of various factors such as gender, age, or education level with immigrant status.

In synthesizing recommendations for improvement, we will categorize suggestions thematically, discussing policy changes, service delivery modifications, and cultural competency training, among others. We will highlight frequently cited recommendations as well as those that are particularly innovative or evidence based. Our discussion will include an assessment of the feasibility and potential impact of key recommendations.

Based on our synthesis of findings, we will develop and present a conceptual framework illustrating the key factors influencing immigrants' access to and experiences with mental health services and psychosocial supports in Canada. This framework will visually represent the interrelationships between experiences, barriers, enablers, and recommendations, accompanied by a detailed narrative explanation.

Our analysis will also critically examine gaps in the current literature, including underrepresented immigrant populations or geographic areas, understudied types of mental health services or interventions, and methodological limitations in existing research.

Finally, we will provide a balanced assessment of the strengths and limitations of both the included articles and our scoping review methodology. This will include a discussion of potential biases in the literature and our review process, as well as the implications of these factors for interpreting and applying the findings.

## Discussion

This scoping review maps the literature on immigrants' experiences accessing mental health services and psychosocial support in Canada. It aims to highlight key challenges and factors, including the "healthy immigrant effect," where immigrants' health, including mental health, declines over time [1,2,6,29]. We expect our findings to synthesize literature on barriers

immigrants face in accessing mental health care, including language, cultural differences, stigma, lack of awareness, and systemic discrimination [1,2,9,10]. Our review may identify how barriers vary across subgroups or regions and how factors like gender or socioeconomic status interact with immigrant status. It will also highlight enablers and strategies to improve access, such as culturally adapted interventions, language-concordant care, and community-based support. These findings could guide best practices and policy development, aligning with the Mental Health Commission's strategic directions, and reveal emerging trends to improve access and outcomes for immigrant populations [30].

This review's focus on Canada's unique healthcare system and immigrant demographics allows for targeted recommendations, though findings may not be generalizable to other countries. Future research could compare different national contexts. The inclusion of both quantitative and qualitative studies offers a comprehensive view of immigrants' experiences, aligning with calls for culturally sensitive research in immigrant mental health. (15,16). It may also help bridge the gap between statistical trends and individual narratives, providing a more holistic picture of the immigrant experience with mental health services.

Our findings on the experiences of immigrant families navigating mental health services may provide insights into the cascading effects of mental health challenges on education, employment, and social integration (11,12). This review may highlight the need for early intervention and support beyond clinical settings, emphasizing school-based programs, community outreach, and family-centered approaches for immigrant populations. It could also identify gaps in research, such as a lack of studies on specific subgroups or mental health conditions, guiding future priorities and promoting participatory research.

The conceptual framework developed could guide multi-level interventions, though further research is needed to validate it. Publication bias and the focus on peer-reviewed English literature in Canada may limit the range of perspectives. In conclusion, this review aims to advance understanding of immigrant experiences with mental health services in Canada, inform policy, practice, and research, and contribute to a more equitable mental health system for all Canadians.

## Supporting Information

**S1 Table. This is the S1 Table 1 Prisma-P.**
(DOCX)

**S2 Table. This is the S2Information Draft Strategy for Medline.**
(DOCX)

**S3 File. This is the draft search strategy.**
(DOCX)

## Acknowledgments

The authors would like to thank Melissa Rothfus, Health Sciences Librarian at Dalhousie University's Kellogg Library, for her invaluable assistance in developing the search strategy for this study. Her expertise and guidance were instrumental in ensuring a comprehensive and rigorous approach to the literature search.

## Author contributions

**Conceptualization:** Temitayo Sodunke, Belinda Agyapong, Mutiat Sulyman, Sanjana Sridharan, Ejemai Eboreime.

**Data curation:** Gerald Agyapong-Opoku.

**Formal analysis:** Omolayo Anjorin.

**Project administration:** Somtoo Rita Henry.

**Supervision:** Ejemai Eboreime.

**Validation:** Mutiat Sulyman, Sanjana Sridharan, Ejemai Eboreime.

**Visualization:** Ejemai Eboreime.

**Writing – original draft:** Temitayo Sodunke.

**Writing – review & editing:** Temitayo Sodunke, Belinda Agyapong, Ejemai Eboreime.

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
