## [Decision Letter · Decision Letter 0]

13 Jan 2025

PONE-D-24-44279Towards Equitable Mental Health Care: A Scoping Review Protocol on Immigrant Experiences in CanadaPLOS ONE

Dear Dr. Eboreime,

Thank you for submitting your manuscript to PLOS ONE. After careful consideration, we feel that it has merit but does not fully meet PLOS ONE’s publication criteria as it currently stands. Therefore, we invite you to submit a revised version of the manuscript that addresses the points raised during the review process.

We look forward to receiving your revised manuscript.

Kind regards,

Nancy Clark, PhD

Academic Editor

PLOS ONE

Journal Requirements:

This work was funded by a QE II Foundation Translating Research into Care (TRIC) grant awarded to EE (Grant number 1029792). The funders played no role in the study design, decisions to publish or preparation of the manuscript.

Eboreime E. 

This work was funded by the QEII Foundation Translating Research into Care (TRIC) Award 1029792. The funders played no role in the study design, data collection, data extraction, results, and discussion.

https://qe2foundation.ca/qeii-health-sciences-centre-staff/grants-health-centre-staff-physicians/translating-research-care-tric-grant-program

Additional Editor Comments :

Dear authors, we have received all required reviewers, and we recommend major revisions to address reviewer's comments. Please note that if there are comments you disagree with, please provide a rationale. We invite you to re submit your manuscript once you have attended to the recommendations.

There are some additional comments I would like to make regarding updating references.

1. For example, reference to mental health commission of Canada strategies see 2016 vs 2009 line 71. https://www.mentalhealthcommission.ca/wp-content/uploads/drupal/2016-10/case_for_diversity_oct_2016_eng.pdf

2. Please also clarify what is meant related to intergenerational stress throughout you make reference to youth however is the scoping review focused on all ages or youth only?

3. Also, if the scope of the review is to understand immigrant's experiences of mental health services and address their needs; would you consider explaining if and why people with lived experience are or are not included in the review? I would recommend explaining also inclusion criteria related to who are newcomers? e.g. are refugees included as a subcategory see line 133, because their experiences maybe different in terms of equity?

4. The overall focus on experience, it is not clear why quantitative studies will also be included unless you are focusing on structural issues related to health outcomes? This could be addressed in the ethics section.

5. I appreciate the inclusion of inter-rater reliability however it is a quantitative method you may want to consider intercoder ICR reliability but that is up to you line 158 see article by O'Connor: https://journals.sagepub.com/doi/full/10.1177/1609406919899220

6. p.9 Table 1 there is nothing listed under mental health characteristics? do you mean mental health problems based on DSM V criteria? you may wish to exclude this as you already have a category listed below this one?

Integration of findings should relate to equity centred design or approach to analysis? - analysis should be equity focused.

7. Re healthy immigrant effect, research has also documented same effects for mental health over time for immigrants you want to include a reference to this https://www.researchgate.net/publication/325338568_Canadian_Immigrant_Mental_Health and or other references related to mental health decline specifically.

Again, thank you for this important work, I hope the reviewer's comments and these recommendations will be helpful in your work,

We look forward in receiving your revised manuscript,

Kindest regards,

Nancy Clark PhD

PLOS ONE

Reviewers' comments:

Reviewer's Responses to Questions

**Comments to the Author**

1. Does the manuscript provide a valid rationale for the proposed study, with clearly identified and justified research questions?

Reviewer #1: Partly

Reviewer #2: Partly

2. Is the protocol technically sound and planned in a manner that will lead to a meaningful outcome and allow testing the stated hypotheses?

Reviewer #1: Partly

Reviewer #2: Partly

3. Is the methodology feasible and described in sufficient detail to allow the work to be replicable?

Reviewer #1: No

Reviewer #2: Yes

4. Have the authors described where all data underlying the findings will be made available when the study is complete?

Reviewer #1: Yes

Reviewer #2: No

5. Is the manuscript presented in an intelligible fashion and written in standard English?

Reviewer #1: Yes

Reviewer #2: Yes

6. Review Comments to the Author

You may also provide optional suggestions and comments to authors that they might find helpful in planning their study.

Reviewer #1: Dear authors,

Thank you for the opportunity to review the manuscript #PONE-D-24-44279. I agreed to review this scoping review protocol as it has a focus on mental health care for immigrant communities in Canada.

I would like to commend the reviewers on this very important work and congratulations on receiving funding.

Please see my recommendations below.

Abstract:

- Please clarify if you are focusing on immigrants or newcomers. Please make this clear throughout the protocol.

- For example, you have stated, “There is a dearth of articles around newcomer experiences while accessing mental health and psychosocial support which has necessitated our current study.”, but then focus on “immigrants”.

- This sentence is not clear: “Therefore, to better address the needs of a diverse immigrant population, we are conducting a scoping review is to map out the scope of literature available on the experiences of immigrants while accessing mental health services and psychosocial support in Canada.” While the study appears to be sound, the language at times is unclear, making it difficult to follow. I advise the authors work with a writing coach or copyeditor to improve the flow and readability of the text.

- I am not sure references are used in abstract. I will defer this to the editor’s decision.

Introduction:

Please provide a reference for the following:

- “healthy immigrant effect”

- Not clear on “diverse immigrant population” is meant. Please clarify your definition of diversity for the reader.

- The word ‘migrant’ is introduced here. Please see my suggestion above in clarifying terminology for the reader.

- Although important, this is a very broad topic. There is extensive diversity within the immigrant communities. I would suggest a much narrower focus. Are you focusing on a category of immigrants? Within that category, are you focusing on a particular group (e.g. ethnicity, age, religion, gender identity and expression, sexual orientation, education, those who escaped war, etc.)?

Review question:

- The word “experience” lend itself to a qualitative systematic review

Method:

- Please clarify the rational for not using a more up to date scoping review approaches (e.g. JBI) compared to Arksey and O'Malley's

Search strategy:

- You have stated, “search strategy will be developed in consultation with a health sciences librarian…” Has this librarian been involved in the methods of this scoping review protocol? Please clarify who developed the search strategy in Appendix B.

- Please provide references to support the limitation for the last 10 years

- Are you selecting studies or articles? At times you have said articles and other times you have stated studies. Please clarify this for the reader. Would you only be selecting research studies or go beyond studies? What about grey literature? What about relevant websites and etc.

- As this is a Scoping review, please provide a rational for only selecting peer-reviewed journal articles.

- The reason for excluding the following for a scoping review is not clear: “Case reports, systematic reviews, meta-analyses, interventional studies, commentaries, editorials, opinion pieces, and grey literature”

- Please provide a detail plan for pilot testing of each stage of review and data extraction. E.g. how many will be pilot tested for each stage, what inter-rater reliability will you consider satisfactory?

Data extraction:

- Please add “Notes” and “relevant articles from the reference list” for your data extraction table

Data analysis and results:

- Usually “thematic analysis” is not used in SR because we are not interpreting the extracted data. This would fall within qualitative Systematic review. I would suggest the team to consider content analysis. There is a newly published scoping review by Limoges et al. that you may find extremely helpful in your data analysis approach.

- You may find these helpful: https://journals.plos.org/plosone/article?id=10.1371/journal.pone.0295914

- Limoges J, Chiu P, Dordunoo D, Puddester R, Pike A, Wonsiak T, Zakher B, Carlsson L, Mussell JK. Nursing strategies to address health disparities in genomics-informed care: a scoping review. JBI Evid Synth. 2024 Sep 11;22(11):2267–312. doi: 10.11124/JBIES-24-00009. Epub ahead of print. PMID: 39258479; PMCID: PMC11554251.

Discussion:

- It is not clear how this will be ‘comprehensive’ with a limited focused on peer-reviewed research articles.

- There is a lot of information in the discussion section. Please note that you cannot take anything verbatim from your protocol into your paper that will be focused on the report of scoping review. I would suggest the authors to consider condensing this section.

Timeline:

- I am not clear on the timeline section. I think it can be removed, but I will leave it for the editor’s decision.

I hope you find this feedback helpful as you continue publishing your work.

Reviewer #2: General comments

Check for grammar throughout the manuscript

Specific comments

Title: the idea of equitable as used in the title is not explicit in the body of the manuscript. Is it assumed that immigrants don’t have equitable access to mental health services? If so, that should be evident in the text. It should be argued in the background section.

Introduction

The intext reference starts at [2]. Please indicate reference for [1].

Page 3 line 55- give example of these incidences and elaborate on how they compare with the general population

Line 61- healthy immigrant effect is often applied to physical health. Please provide a brief writeup on how immigrants have better mental health status at the time of migration and worsen with the length of stay, to demonstrate how this concept is relevant to mental health.

Line 63- there are other factors that determine the access to services that deserve mention. Perhaps doing a thorough review will unearth them- including cultural competency of providers, mental health literacy, stigma, misconception, differences in the way mental health is conceptualised between home and host country, differences in culture etc.

Substantiate all the statements you make- like in line 66.

Page 4 line 71- more information about mental health strategy and if it has any reference to immigrant mental health is warranted.

Line 74- too long and would read better if divided into two

Line 77- statement on comprehension curriculum is stigmatising and blaming the student and prejudicial. Consider changing the language to different learning style and approaches.

Line 80- clarify what you mean with inadequate accessibility, and statement as expressed by other reviewers- do you mean scholars or researchers?

Line 88 clarify what one size fits all approach looks like

Page 5- a justification statement is required on what necessitated the study. A case study drawn from a clinical experience with immigrants would suffice. Also, if authors are themselves immigrants, they can draw from their own lived experiences.

Line 95- clinical significance of this work is vital.

Research questions are broad and need to be revised and made more focussed

Focussing on Canada alone may not give adequate materials to work with. In addition, you need to make a justification as to why you are limiting to Canada and not traditionally to other immigrant accepting countries.

Page 7- line 139. Why limit the search to 10 years?

Table 1- broad and are at risk of not finding the information you are looking for if you intend to populate each section. A typical data extraction sheet usually has 8 columns. Might be better to familiarise yourself with a similar scoping review and learn what information ends to be extracted for analysis.

Line 189- develop the section on analysis further

Line 195- what will be the component of the conceptual framework? This section needs to be developed further.

Line 204- be specific on who the stakeholders are and where the consultation will be and how it will be conducted.

Results- line 266- not necessary for a protocol as there are none to report.

Discussion- should be summarised and focus on what the scoping review will achieve especially on implication for clinical practice. Two paragraph are adequate.

7. PLOS authors have the option to publish the peer review history of their article (what does this mean? ). If published, this will include your full peer review and any attached files.

**Do you want your identity to be public for this peer review?** For information about this choice, including consent withdrawal, please see our Privacy Policy .

Reviewer #1: No

Reviewer #2: No

---

## [Author Response · Author response to Decision Letter 1]

29 Jan 2025

RESPONSES TO REVIEWRS

Thank you for this we have revised the manuscript according to the style requirements

Thank you for stating the following in the Acknowledgments Section of your manuscript:

This work was funded by a QE II Foundation Translating Research into Care (TRIC) grant awarded to EE (Grant number 1029792). The funders played no role in the study design, decisions to publish or preparation of the manuscript. We note that you have provided funding information that is not currently declared in your Funding Statement. However, funding information should not appear in the Acknowledgments section or other areas of your manuscript. We will only publish funding information present in the Funding Statement section of the online submission form. Please remove any funding-related text from the manuscript and let us know how you would like to update your Funding Statement. Currently, your Funding Statement reads as follows: Eboreime E. This work was funded by the QEII Foundation Translating Research into Care (TRIC) Award 1029792. The funders played no role in the study design, data collection, data extraction, results, and discussion.

https://qe2foundation.ca/qeii-health-sciences-centre-staff/grants-health-centre-staff-physicians/translating-research-care-tric-grant-program

Funding: This scoping review is funded by the QE II Foundation Translating Research into Care (TRIC) grant. Please include your amended statements within your cover letter; we will change the online submission form on your behalf.

We have removed the funding statement from the acknowledgement section, and the statement has not changed, so it can remain the same as it is on the online submission

Please provide a complete Data Availability Statement in the submission form, ensuring you include all necessary access information or a reason for why you are unable to make your data freely accessible. If your research concerns only data provided within your submission, please write "All data are in the manuscript and/or supporting information files" as your Data Availability Statement.

This has been included in the submission form. Given that this is a protocol for a study yet to be conducted, we currently have no data to share.

Please include captions for your Supporting Information files at the end of your manuscript, and update any in-text citations to match accordingly. Please see our Supporting Information guidelines for more information: http://journals.plos.org/plosone/s/supporting-information.

A caption for the table, captioned S1Table Draft search strategy for MEDLINE has been included as part of the supporting information

Additional Editor Comments:

For example, reference to mental health commission of Canada strategies see 2016 vs 2009 line 71. https://www.mentalhealthcommission.ca/wp-content/uploads/drupal/2016-10/case_for_diversity_oct_2016_eng.pdf

Thank you for sharing the more recent strategy. We have replaced the 2009 reference with the 2016 strategy and updated the information accordingly. Please see line 79 for the revised section.

Please also clarify what is meant related to intergenerational stress throughout you make reference to youth however is the scoping review focused on all ages or youth only?

Thank you for your comment. The paper is focused on first-generation immigrants of all ages, not specifically on youth. This has been consistently reflected throughout the paper. Additionally, the term "intergenerational stress" is no longer used in the paper.

Also, if the scope of the review is to understand immigrant's experiences of mental health services and address their needs; would you consider explaining if and why people with lived experience are or are not included in the review? I would recommend explaining also inclusion criteria related to who are newcomers? e.g. are refugees included as a subcategory see line 133, because their experiences maybe different in terms of equity?

Thank you for your suggestion. We have clarified that our population includes first-generation immigrants, which encompasses newcomers such as refugees, whose experiences may differ in terms of equity. We have expanded the explanation based on IRCC guidelines (see lines 158 to 162. As this is a scoping review, depending on the types of papers extracted, it could potentially include individuals with lived experiences.

The overall focus on experience, it is not clear why quantitative studies will also be included unless you are focusing on structural issues related to health outcomes? This could be addressed in the ethics section.

Thank you for your comment. Depending on the types of papers that meet our inclusion criteria during the extraction process, quantitative studies will also be included, especially if they address and measure relevant factors such as structural issues related to health outcomes. This approach is in line with the scoping review guidelines. It is unclear why this should be an ethical issue, particularly as review articles typically do not require ethics approvals

I appreciate the inclusion of inter-rater reliability however it is a quantitative method you may want to consider intercoder ICR reliability but that is up to you line 158 see article by O'Connor: https://journals.sagepub.com/doi/full/10.1177/1609406919899220

Thank you for your advice. We have changed to refect intercoder ICR reliability.

p.9 Table 1 there is nothing listed under mental health characteristics? do you mean mental health problems based on DSM V criteria? you may wish to exclude this as you already have a category listed below this one?

Thank you for this, we have removed the row, and left the row for mental health conditions as they are representing the same things, see Table 1 from the supporting information

Integration of findings should relate to equity centered design or approach to analysis? - analysis should be equity focused.

Thank you for your suggestion. We have rewritten the section on the integration of findings to emphasize how the paper adopts an equity-focused approach. Please see line 225 for the updated content.

Re healthy immigrant effect, research has also documented same effects for mental health over time for immigrants you want to include a reference to this https://www.researchgate.net/publication/325338568_Canadian_Immigrant_Mental_Health and or other references related to mental health decline specifically.

Thank you for your comment. This has been addressed, and references to the mental health effects, including the suggested reference, have been added. Please see lines 53 to 61 for the updated content.

Reviewer #1:

Abstract:

Please clarify if you are focusing on immigrants or newcomers. Please make this clear throughout the protocol.For example, you have stated, “There is a dearth of articles around newcomer experiences while accessing mental health and psychosocial support which has necessitated our current study.”, but then focus on “immigrants”.

Thank you for pointing this out. While we initially used the term “newcomers,” we have clarified that our focus is on immigrants in general, particularly first-generation immigrants, which includes newcomers. To ensure clarity and consistency, we have removed the term “newcomers” and specified that the study encompasses all immigrants, as reflected in line 25.

This sentence is not clear: “Therefore, to better address the needs of a diverse immigrant population, we are conducting a scoping review is to map out the scope of literature available on the experiences of immigrants while accessing mental health services and psychosocial support in Canada.” While the study appears to be sound, the language at times is unclear, making it difficult to follow. I advise the authors work with a writing coach or copyeditor to improve the flow and readability of the text.

This sentence has been improved to ensure readability and clarity see line 28

I am not sure references are used in abstract. I will defer this to the editor’s decision.

The reference has been taken out

Introduction:

Please provide a reference for the following:

- “healthy immigrant effect”

Thank you for your comment. This has been addressed, and we have included a broad definition and references for the “healthy immigrant effect”. Please see lines 54 to 58 for the updated content.

Not clear on “diverse immigrant population” is meant. Please clarify your definition of diversity for the reader.

"Diverse immigrant population" was unclear and has been revised to "immigrant population" throughout the document for accuracy and clarity.

The word ‘migrant’ is introduced here. Please see my suggestion above in clarifying terminology for the reader.

This has been changed across the document to immigrant, to maintain consistency of terminology

Although important, this is a very broad topic. There is extensive diversity within the immigrant communities. I would suggest a much narrower focus. Are you focusing on a category of immigrants? Within that category, are you focusing on a particular group (e.g. ethnicity, age, religion, gender identity and expression, sexual orientation, education, those who escaped war, etc.)?

Thank you for this and we agree it is a broad category, which is why we are focusing on first generational immigrants. Considering that this is a scoping review, the focus is typically broad and exploratory, in contrast with a systematic review. The purpose is to map evidence towards future more focused research and/or policy.

Review question:

The word “experience” lend itself to a qualitative systematic review

The term “experience” is appropriate for a qualitative systematic review; however, scoping reviews can also explore experiences as well. For reference, see Mak, S., & Thomas, A. (2022). Steps for Conducting a Scoping Review. Journal of Graduate Medical Education, 14(5), 565–567. https://doi.org/10.4300/JGME-D-22-00621.1. Further, we are not excluding quantitative articles from our review considering that this is a scoping review and there is the possibility that some articles may apply quantitative measures to experiences, which we do not want to miss out. For example, a similar study in Europe found 5 quantitative vs 3 qualitative articles on immigrant experiences. For reference see Kjøllesdal, M.K.R., Iversen, H.H., Skudal, K.E. et al. Immigrant and ethnic minority patients` reported experiences in psychiatric care in Europe – a scoping review. BMC Health Serv Res 23, 1281 (2023). https://doi.org/10.1186/s12913-023-10312-1

Method:

Please clarify the rational for not using a more up to date scoping review approaches (e.g. JBI) compared to Arksey and O'Malley's

Arksey and O'Malley's framework is a widely recognized and established method for conducting scoping reviews. We have chosen to complement this approach with the JBI methodology to enhance the review process. Please see line 135 for further details.

Search strategy:

You have stated, “search strategy will be developed in consultation with a health sciences librarian…” Has this librarian been involved in the methods of this scoping review protocol? Please clarify who developed the search strategy in Appendix B.

The search strategy attached was done in consultation with a health sciences librarian at Dalhousie University’s Kellog Library (please see acknowledgement).

Please provide references to support the limitation for the last 10 yea

We have removed the specified time frame.

Are you selecting studies or articles? At times you have said articles and other times you have stated studies.

We have reflected this to include one uniform terminology that is articles as they represent a broad range of papers. This shown throughout the paper

Please clarify this for the reader. Would you only be selecting research studies or go beyond studies? What about grey literature? What about relevant websites and etc.

We have expanded our inclusion criteria to include grey literature please see line 165

As this is a Scoping review, please provide a rational for only selecting peer-reviewed journal articles.

We have expanded our inclusion criteria to include grey literature and case reviews please see line 165

The reason for excluding the following for a scoping review is not clear: “Case reports, systematic reviews, meta-analyses, interventional studies, commentaries, editorials, opinion pieces, and grey literature”

Thank you for the suggestion. We have clarified our inclusion criteria. Grey literature, including case reports, will now be included. However, as this is not a scoping review of systematic reviews, we will still exclude those. Please see line 171 for the updated information.

Please provide a detail plan for pilot testing of each stage of review and data extraction. E.g. how many will be pilot tested for each stage, what inter-rater reliability will you consider satisfactory?

Thank you for the suggestion. We have included a detailed plan for pilot testing, specifying that we will pilot test 50 titles and abstracts. We will also use inter-rater reliability as a measure, with the acceptable level outlined in the revised section. Please see lines 184 to 187 for the details.

Data extraction:

Please add “Notes” and “relevant articles from the reference list” for your data extraction table

Data analysis and results:

Thank you for this suggestion, we have included this on our Draft search strategy table

Usually “thematic analysis” is not used in SR because we are not interpreting the extracted data. This would fall within qualitative Systematic review. I would suggest the team to consider content analysis. There is a newly published scoping review by Limoges et al. that you may find extremely helpful in your data analysis approach.

- You may find these helpful: https://journals.plos.org/plosone/article?id=10.1371/journal.pone.0295914

- Limoges J, Chiu P, Dordunoo D, Puddester R, Pike A, Wonsiak T, Zakher B, Carlsson L, Mussell JK. Nursing strategies to address health disparities in genomics-informed care: a scoping review. JBI Evid Synth. 2024 Sep 11;22(11):2267–312. doi: 10.11124/JBIES-24-00009. Epub ahead of print. PMID: 39258479; PMCID: PMC11554251.

Thank you for this. However, thematic analysis can be used in qualitative research. Please see Mak, S., & Thomas, A. (2022). Steps for Conducting a Scoping Review. Journal of Graduate Medical Education, 14(5), 565–567. https://doi.org/10.4300/JGME-D-22-00621.1. We have also referenced its use in the manuscript—please see line 218.

Discussion:

It is not clear how this will be ‘comprehensive’ with a limited focused on peer-reviewed research articles.

This has been removed

There is a lot of information in the discussion section. Please note that you cannot take anything verbatim from your protocol into your paper that will be focused on the report of scoping review. I would suggest the authors to consider condensing this section.

We have summarized this to include the main points

Timeline:

I am not clear on the timeline section. I think it can be removed, but I will leave it for the editor’s decision.

This was taken out

I hope you find this feedback helpful as you continue publishing your work.

Reviewer #2: General comments

Check for grammar throughout the manuscript

Thank you.

Specific comments

Title: the idea of equitable as used in the title is not explicit in the body of the manuscript. Is it assumed that immigrants don’t have equitable access to mental health services? If so, that should be evident in the text. It should be argued in the background section

The title refers to the experiences of immigrants, in accessing health care and does not talk about equity.

Introduction

The intext reference starts at [2]. Please indicate reference for [1].

The in-text references now start at [1]. The reference previously in the abstract has been removed, and

---

## [Editor Report · Decision Letter 1]

7 Feb 2025

Experiences of immigrants when assessing mental health services and psychosocial support in Canada: Protocol for a scoping review

PONE-D-24-44279R1

Dear Dr. Eboreime

We’re pleased to inform you that your manuscript has been judged scientifically suitable for publication and will be formally accepted for publication once it meets all outstanding technical requirements.

Kind regards,

Nancy Clark, PhD

Academic Editor

PLOS ONE

Additional Editor Comments (optional):

Thank you for your updated and detailed response to the reviewers that were integrated into the manuscript. I have read the updated manuscript and your response to reviewers and find it satisfactory for publication.
---

## [Editor Report · Acceptance letter]

PONE-D-24-44279R1

PLOS ONE

Dear Dr. Eboreime,

I'm pleased to inform you that your manuscript has been deemed suitable for publication in PLOS ONE. Congratulations! Your manuscript is now being handed over to our production team.

Kind regards,

on behalf of

Dr. Nancy Clark

Academic Editor

PLOS ONE